# How Can Biomechanics Improve Physical Preparation and Performance in Paralympic Athletes? A Narrative Review

**DOI:** 10.3390/sports9070089

**Published:** 2021-06-24

**Authors:** Jared R. Fletcher, Tessa Gallinger, Francois Prince

**Affiliations:** 1Department of Health and Physical Education, Mount Royal University, Calgary, AB T3E 6K6, Canada; 2Canadian Sport Institute Calgary, Calgary, AB T3B 6B7, Canada; tgallinger@csicalgary.ca; 3Department of Surgery, Faculty of Medicine, University of Montreal, Montreal, QC H3T 1J4, Canada; francois.prince@umontreal.ca; 4Institut National du Sport du Québec, Montréal, QC H1V 3N7, Canada

**Keywords:** kinematics, prostheses, classification, amputee, cerebral palsy

## Abstract

Recent research in Paralympic biomechanics has offered opportunities for coaches, athletes, and sports practitioners to optimize training and performance, and recent systematic reviews have served to summarize the state of the evidence connecting biomechanics to Paralympic performance. This narrative review serves to provide a comprehensive and critical evaluation of the evidence related to biomechanics and Paralympic performance published since 2016. The main themes within this review focus on sport-specific body posture: the standing, sitting, and horizontal positions of current summer Paralympic sports. For standing sports, sprint and jump mechanics were assessed in athletes with cerebral palsy and in lower-limb amputee athletes using running-specific prostheses. Our findings suggest that running and jumping-specific prostheses should be ‘tuned’ to each athlete depending on specific event demands to optimize performance. Standing sports were also inclusive to athletes with visual impairments. Sitting sports comprise of athletes performing on a bike, in a wheelchair (WC), or in a boat. WC configuration is deemed an important consideration for injury prevention, mobility, and performance. Other sitting sports like hand-cycling, rowing, and canoeing/kayaking should focus on specific sitting positions (e.g., arm-crank position, grip, or seat configuration) and ways to reduce aero/hydrodynamic drag. Para-swimming practitioners should consider athlete-specific impairments, including asymmetrical anthropometrics, on the swim-start and free-swim velocities, with special considerations for drag factors. Taken together, we provide practitioners working in Paralympic sport with specific considerations on disability and event-specific training modalities and equipment configurations to optimize performance from a biomechanical perspective.

## 1. Introduction

In 2016, 176 countries and more than 4000 athletes competed at the Paralympic Games. These athletes remain significantly understudied compared to Olympic athletes, especially with regards to the role the field of biomechanics can serve in improving physical preparation and performance. Recently, Morriën et al. [1] conducted a systematic review of biomechanical studies in Paralympic research consisting of 41 articles published before July 2016, showing that the majority of the included studies contribute to our understanding of technical optimization, injury prevention, and evidence-based classification. Because of the nature of the systematic review itself, a critical analysis of the 41 biomechanical studies was not included. Here, we serve to update this review and to examine the impact that specific biomechanical interventions may have on Paralympic performance. We construct this narrative review by considering Paralympic athletes and their specific impairment(s) and their sport’s specific body posture: standing, sitting, and horizontal positions (Table 1). We summarize the newest biomechanical evidence related to these athletes and their physical preparation, performance, and potential use of technological innovation. We then offer practical suggestions for coaches, Paralympic athletes, and sports practitioners to optimize Paralympic athlete performance.

## 2. Standing

Standing postures are part of many Paralympic events (Table 1). We focus our narrative on the main themes of biomechanics research published since 2016: sprinting and jumping with prostheses, sprinters with cerebral palsy, and considerations for athletes with a visual impairment. We recognize that other standing postures may be omitted here given the lack of biomechanics-related research published since 2016.

### 2.1. Sprinting Biomechanics: The Basics

Running speed is the product of step length and step frequency. These changes in kinematics are produced as the result of changes in kinetic quantities. Step length increases by applying greater horizontal and vertical ground reaction forces during ground contact, which increases a runner’s horizontal and vertical takeoff velocities and thus increases the aerial time and the horizontal distance covered by the runner’s centre of mass [2,3]. Step frequency increases are achieved by reducing foot–ground contact time and/or the time taken to reposition the swinging limb for the next step.

### 2.2. Amputee Mechanics

Unilateral and bilateral lower limb amputee runners use running-specific prostheses made of carbon-fibre sockets and blades attached in-series with the residual limb(s). The blade works as a leaf spring to store and release elastic energy during running [4], returning up to 95% of the mechanical energy stored in them [5]. Unlike running-specific prostheses, lower limbs ‘return’ greater than 100% mechanical energy since active muscle contraction contributes to positive joint work [6]; thus, compared to an intact lower limb, running-specific prostheses are disadvantaged with regard to energy storage and return. The storage and return of the mechanical work in intact human limbs require metabolic energy through active muscle contraction [7], which is not the case with running-specific prostheses. Running-specific prostheses also do not allow any neural adjustments (from active muscle contraction) in stiffness during running. Thus, stark differences in lower limb kinematics between able-bodied and lower-limb amputee runners have been reported [8,9,10,11,12,13]. Running-specific prostheses are also available in different models with different geometries [14] and mechanical properties [5], and they can be modified for running-specific prosthetic-socket alignment and prosthetic height [15], the latter of which is to respect the maximum allowable standing height as regulated by the International Paralympic Committee.

The running kinematics and kinetics of elite Paralympians using running-specific prostheses and whether these devices incur an advantage to the Paralympians has been contentious [16,17]. Unilateral transtibial lower-limb amputees achieve top speeds by eliciting different spatiotemporal and vertical GRF characteristics compared to bilateral lower-limb amputees or able-bodied sprinters [9]. Specifically, sprinting with running-specific prostheses results in longer step lengths compared to a biological limb as well as lower and longer vertical GRF compared to the sound leg [9,10]. Thus, similar vertical impulses can be produced with lower peak vertical GRF forces with running-specific prostheses; the total resultant force application does not need to be as high nor the lower limb strength as great in sprinters using running-specific prostheses.

Running-specific prosthetic height and stiffness, the latter of which is influenced by running speed, can be modified. Taboga et al. [15] found, however, no effect of height or stiffness on maximum sprint speed (over a relatively narrow range of stiffnesses (±1 stiffness category) or heights (±2 cm)). This does not imply that changes in stiffness and/or height beyond those that were tested would not infer a performance advantage. Indeed, a recent ruling by the Court of Arbitration for Sport ruled a Paralympic Champion ineligible to compete at the Olympics largely as a result of his running-specific prosthetic height [18].

Running-specific prosthetic alignment in the sagittal plane relative to the intact limb has only recently been investigated. Migliore et al. [19] examined the impact of increasing the sagittal tilt from 5° to 12° relative to the line of gravity in a gold medal Paralympian. With this change in alignment, the athlete was able to increase step frequency and reduce step length with the sound side. With a higher sagittal tilt, the propulsive impulse was higher, and the braking impulse decreased. Importantly, the athlete perceived the increased tilt as the best running-specific prosthetic alignment, and shortly after implementing the tilt alignment, she improved upon her personal best in the T42 100 m, narrowly missing the world record.

Importantly, athletes looking to change their running-specific prosthetic model, alignment, and/or height must be aware that these changes likely alter the stiffness profile of the running-specific prosthetic [5]. Specifically, the running-specific prosthetic stiffness increases with the magnitude of the peak resultant GRF but decreases when the angle between the running-specific prosthetic and resultant GRF is increased. This inverse relationship between GRFs and angle may explain previous reports that running-specific prosthetic stiffness does not change during running [10,11].

The maximum resultant force an athlete can produce limits the maximal sprint speed around a curve [2,20]. Running around a curve requires athletes to apply an additional centripetal force to change direction in addition to the horizontal and vertical propulsive forces. In sprint events of 200 m and 400 m, an athlete must negotiate either one or two curves, respectively, with the inside leg relative to the curve always being the left leg (since all races are run in the counterclockwise direction). This may disadvantage left legged single lower-limb amputee sprinters since both the compliance and passive nature of running-specific prostheses reduce the application of the maximal forces on the ground [10]. Indeed, lower-limb amputee sprinters were 4% slower with the affected leg on the inside of the curve compared to sprinting with the affected leg on the outside of the curve due to their inability to generate large forces with their affected leg [21]. Single-leg strength and speed training may be required in order to improve the affected limb’s ability to generate large vertical and centripetal forces during sprint running, and/or athletes should consider a better prosthetic design to help improve curve running in single and double-leg amputee sprinters.

### 2.3. Mechanics of Sprinters with Cerebral Palsy

Cerebral palsy is associated with a series of permanent movement disorders resulting from an upper motor neuron lesion in the brain. This impairment is not progressive in nature, but it likely causes secondary adaptations to the muscle structure, function, and composition [22]. Athletes with cerebral palsy will experience muscle weakness [23], increased antagonist coactivation [24], spasticity [25], reduced muscle and joint power, and limited range of motion [26].

Sprinters with cerebral palsy show lower force production compared to their able-bodied counterparts, which is likely a result of the reduced lower-limb strength in cerebral palsy. For example, over a 10 m sprint, a Paralympic sprint medalist with cerebral palsy (T36) showed lower average horizontal power compared to able-bodied sprinters, attributable to both higher horizontal braking and lower propulsive forces resulting in a lower net propulsive impulse [27]. The larger horizontal braking forces in the athlete with cerebral palsy, along with a stiffer ankle joint, were attributed to the increased muscle co-contractions commonly observed in cerebral palsy [28,29,30]. Increased passive and active (from antagonist co-activation) joint stiffness contribute to the reduced joint angular velocities, external joint power, and reductions in step length during the initial acceleration phase of sprinting. 

In summary, training interventions should focus on strategies to increase lower limb joint angular velocities and positive joint powers. Increases in the strength and power of the muscles that cross the hip and knee extensors is recommended to improve sprint performance, primarily because the powerful hip joint performs negative work in cerebral palsy yet performs net positive work in able-bodied athletes [27].

### 2.4. Biomechanics of Jumping

The performance gap between the Olympic and unilateral lower-limb amputee Paralympic winning jumps is reducing (Figure 1). In the past, Paralympic champions have been permitted to compete at the Olympics on the track [13], and it may soon be the case that Paralympic long jumpers will compete at the Olympics as well.

Biomechanically, the goal of the long jump is to maximize the horizontal distance of the jump. This is accomplished by generating as much vertical velocity at take-off while minimizing the loss in horizontal velocity gained on the approach run-up. To do so, Olympic and unilateral lower-limb amputee long jumpers lower their centre of mass on the approach to the board to generate as much vertical impulse (and thus vertical velocity) during take-off as possible. Compared to non-amputee jumpers, elite unilateral lower-limb amputees approach the board slower but achieve similar vertical velocities at take-off [31]. The slower horizontal velocity would imply a disadvantage to unilateral lower-limb amputees compared to Olympic jumpers as a result of their running-specific prostheses. However, energy is stored (a portion of which is returned) in running-specific prostheses that may exceed that of a biological limb. In support, Funken et al. [32] have shown that unilateral lower-limb amputees have a longer compression phase and a greater downward motion of the centre of mass, which would be effective in generating vertical impulse and storing and returning energy within the running-specific prostheses. An optimal running-specific prosthetic stiffness, one that maximizes the storage and return of energy while also maximizing the pivot that translates a portion of the horizontal velocity to vertical velocity at take-off [31] should be investigated. This optimal stiffness may be different than the optimal running-specific prosthetic stiffness for straight sprinting. Furthermore, unilateral lower-limb amputee long jumpers could benefit from specific eccentric strengthening of the hip and knee extensors to increase leg stiffness during the pivot and horizontal velocities.

### 2.5. Athletes with Visual Impairments

Reduced or absent visual input in athletes with a visual impairment results in a greater reliance on the somatosensory and vestibular systems for postural control and orientation. Compared to sighted horizontal jumpers, athletes with a visual impairment have a shorter approach run, maintain or increase their speed in the last few strides of the approach, and have an increased board contact time [33] in an attempt to generate as much vertical impulse as possible, which is easier given their slower horizontal velocities compared to sighted jumpers [33]. 

Changes to postural stability and proprioception with altered vision or a sensory deficit may result in abnormal gait patterns in athletes with a visual impairment. The lack of visual input during sprinting can reduce horizontal sprint speed. By incorporating kinesthetic training (exercises with unstable supports), improvements to neuromuscular control of lower limbs and trunk stability may be an important factor in improving kinesthetic awareness [34] and subsequent sprint speed. Therefore, practitioners working with athletes with a visual impairment should incorporate physical assessments of postural and gait imbalances, incorporate stability and proprioceptive training in sport-specific positions, and implement strength programs focused on maximizing horizontal sprinting velocity.

In summary, practitioners and coaches working with Paralympic athletes competing in a standing posture should consider potential limitations to generating the horizontal and vertical impulses during their respective events in order to maximize horizontal sprint speed (in the case of sprinting and jumping) and to improve take-off mechanics in the horizontal jumps.

## 3. Sitting

In many summer Paralympic sports, athletes perform in a sitting position on a bike, in a wheelchair (WC), in a boat, or on a horse. Here, we discuss the recent development of mobility and performance tests used for different WC configurations associated with WC rugby, tennis, and basketball.

### 3.1. Development of Mobility Performance Tests

For field-based WC sports, the ability to evaluate sport-specific mobility performance will provide an indication to an athlete’s relative weaknesses and strengths, monitor overall progress, or examine recovery from an injury. Since 2016, most of the research focused on the development of different mobility tests and the evaluation of their associated validity and reliability. WC Basketball mobility scores were able to discriminate between male/female and between the national and international levels with excellent test–retest reliability [35], but they were unable to differentiate between low and high levels of disability. In WC Tennis, sprint ability and maneuverability capacity showed high inter-trial reliability, while construct validity was achieved with successful discrimination between junior and international player levels [36]. To improve classification objectivity in boccia, dos Santos et al. [37] proposed a spasticity test with rapid passive movements to improve objective classification across para-athletes with cerebral palsy. These tests allow practitioners to objectively quantify and monitor an athlete’s capacity, relative strengths and weaknesses, progress, assess recovery from injury, or improve classification objectivity.

### 3.2. Classification Level, WC Configuration and Mobility Performance

Mobility performance is associated with eight key parameters, including classification level, athlete experience, maximal isometric force, and WC configurations [38]. In WC Basketball, adding a mass to the WC creates a reduction in acceleration, and when distributed, the mass reduces the angular kinematics [39]. These changes were not implemented, but seat elevation enhances sprinting/turning performance while increasing rim grip has a detrimental effect on sprint/turning performance. In WC Rugby, an athlete-specific seat configuration has also been associated with improved sprint performance [40]. As such, differences in propulsion kinematics should be considered for individualized WC configurations and training programs. Since optimum configuration is not always perceived [40], an adequate adaptation period might be needed.

Haydon et al. [41] compared different classification categories in three-stroke acceleration. Sprint performance is reduced in high-disability players from a stationary compared to a sport-specific active start. Surprisingly, this was not seen in the higher-point groups. These results highlight the athlete-specific nature in test design and analyses with these athletes and the need to consider testing modifications across disability groups. Throwing boccia balls, discuses, shot puts, and javelins may require specific WC configurations to enhance performance. Hyde et al. [42] showed that an assistive pole is associated with a higher hand speed in WC throwing. Grip, trunk-flexion strength, and push/pull synergy were also correlated with throwing distance. The use of an assistive pole should definitely be considered in all WC throwing events.

For athletes who are lower-limb amputees, changing the seated configuration will change the body centre of mass position, and the use of prostheses may improve the leverage that the limb(s) can provide. For example, in para kayak, Ellis et al. [43] found that when the prosthetic limb was removed, decrements in stroke rate, stroke speed, stroke length, and overall power output were observed. Coaches should ensure the residual limb is well supported in the boat to provide the maximum leverage for optimal propulsive strength. Adjusting the mass distribution in both the frontal and sagittal planes will also favor buoyancy and reduce hydrodynamic drag. In the case of bilateral lower-limb amputees using a canoe or kayak, we recommend using both lower limb prostheses to distribute the mass evenly and to provide adequate leverage to apply forces from the trunk and upper limbs for efficient paddling.

### 3.3. Aerodynamic Improvements and Performance Enhancement

In addition to optimizing the performance factors of the athlete through physical training, improving the equipment can play a decisive role in enhancing performance in many parasports. Propulsion of a WC, bike, or boat is limited by different sources of resistance including air/water, rolling, gravity, and mechanical friction. Paralympic athletes performing on a bike/WC experience important resistance from aerodynamic drag while Paralympic sports performed in a boat encounter mainly hydrodynamic drag. The air resistance is the most important WC resistive force, especially at higher speeds. Aero/hydrodynamic resistance as well as the energy expenditure can be reduced by adopting a body/bike/boat/WC configuration with minimal frontal area, more aero/hydrodynamic shape, and less surface friction [44]. Using computational fluid dynamics, a lighter prosthetic design with 1 kg reduction and a special body position were associated with a performance improvement of 23 s over a 16.1 km time trial in individual [45,46] and tandem para-cycling [46]. This lower drag and energy cost in amputee cyclists compared to able-bodied cyclists is not present when the model removed one arm or one leg [46]. In para-athletes using hand cycles, Mannion et al. [46] tested different arm-crank positions with computational fluid dynamics and showed that the 9 o’clock arm-crank (relative to top-dead centre) produced less aerodynamic drag compared to the 6 o’clock position. In WC sprinting, different postures during the catch, release, and recovery phases have also been assessed using computational fluid dynamics. After wind tunnel validation [47], higher drag during the catch phase and minimal drag during the recovery phase were reported [44]. Adjusting body configuration and equipment will contribute to reduced drag and improved WC and para-cycling performance.

### 3.4. Specific Training/Physical Preparation Modalities and Risk of Injury

In para-cyclists with or without a unilateral lower-limb amputation, Dyer [48] showed no difference in 1 km time trial performance between groups, suggesting that the sound limb of lower-limb amputee athletes has to further compensate for the contralateral loss to achieve similar cycling performances. This may predispose para-cyclists with unilateral lower-limb amputation to musculoskeletal injuries; the athletes may compensate for the lack of contralateral pushing by increasing rectus/biceps femoris and medial gastrocnemius muscle activity during the pulling phase [49]. Childers et al. [50] compared both the prosthetic and sound sides and found gastrocnemius/rectus femoris compensations for controlling the prosthetic socket. They also reported asymmetrical muscular and moment contributions in para-cyclists. Goosey-Tolfrey et al. [51] tested the kinematics of WC Rugby players during a 15 s sprint. Lower disability players reached greater peak speed and peak power output compared to higher disability players. Propulsion asymmetries were observed more often in lower disability players showing a greater demand for the arms during WC propulsion. 

WC Fencing athletes are classified into three categories (A, B, C) with respect to their movement capacity. A shorter lunge time between Category A and Category B fencers [52] supports the role of the external abdominal oblique and latissimus dorsi muscles as effective postural muscles during fencing attacks. In WC Rugby, a 3 vs. 3 training modality is associated with moderate improvements in WC speed and on the number of high-speed events compared to regular game simulation drills [53]. Thus, reducing the number of players on the court should be considered to improve WC speed. Both the kinetic and kinematic variables can affect Paralympic paddling performance, including the ability to move the trunk, pelvis, and legs. Leg/trunk/arms para-kayakers show less rotation/flexion of the trunk/pelvis and less hip, knee, and ankle flexion during paddling, contributing to lower power outputs in comparison to able-bodied kayakers [54]. 

These studies support the importance of specific physical preparation programs for injury prevention and performance enhancement in Paralympic athletes. Physical preparation should take these asymmetries into account. Athletes with minimal contribution from the lower limbs may benefit from the use of functional electrical stimulation protocols during training to assist with knee flexion/extension movements and improve cardiovascular function [55]. However, the current amplitude, duration, and frequency are among the many parameters to consider when using functional electrical stimulation [56] and may not be sufficient to elicit meaningful changes in muscle force or sport performance. It is also important to consider athletes with asymmetric profiles (cerebral palsy, unilateral amputation), as they are likely to achieve different force applications/joint movements between sides. Therefore, we suggest that coaches focus on improving sport-specific joint ROM and (near) symmetrical force application to improve power outputs in kayak performance. Importantly, the physical preparation of all Paralympic athletes should be category (and thus disability) specific to reduce the incidence of injury and optimize performance.

## 4. Horizontal

Two Paralympic sports utilize a horizontal body position, Paralympic Powerlifting and Paralympic Swimming. Paralympic Powerlifting consists of the adapted bench press, with minimal research available outside recommendations for grip width improving muscle recruitment and activation [57], and potentially reducing risk of injury [58]. This is contrary to the Para-swimming literature, which has accumulated a number of timely and relevant articles published since 2016 to evaluate. 

### 4.1. Swimming

Muscular strength and power are key determinants of improved propulsion during start and free-swimming [59]. Para-swimmers may experience a loss in muscle mass, rate of force development, stability, or coordination affecting swim-start performances. Differences in the anthropometric profile of para-swimmers can also affect their ability to produce symmetrical forces between limbs and may alter drag. Coaches working with para-swimmers need to identify individual characteristics influencing propulsion and drag, acknowledging that the swim-start in para-swimmers is influenced by the severity and type of disability [60].

### 4.2. Effect of Impairment on Swim-Start and Free-Swim Velocities

Assessment of four specific measures of the swim-start highlight distinctive priorities for coaches working with para-swimmers: time, distance, velocity, and force. We suggest that the fastest swim start maximizes horizontal range, resulting from high impulse (and thus high velocity) with low block contact time. Not surprisingly, para-swimmers with greater function are capable of producing a better start and faster free swim speeds [38]. In comparison to able-bodied swimmers, para-swimmers exhibit a greater variability in the start execution, with some adopting compensational mechanisms to help deliver a stable performance. Athletes with neurological impairments tend to show a lower consistency in start performance [61], needing intensive and repetitive practice to develop well-coordinated muscular patterns and increase maximal voluntary activation [62]. 

Free swim speed is critical to overall performance regardless of disability, as it accounts for 67–75% of the variation in 50 m swim performance [60]. This proportion becomes greater the longer the race. Lower velocities during block and underwater phases are associated with a slower time to 15 m, specifically affecting swimmers with lower body or high-severity disabilities who spend a smaller percentage of time in the underwater phase [60]. Swimmers with a visual impairment may lose their orientation or ability to define their position in the water, resulting in significantly slower free-swim speeds [63].

### 4.3. Asymmetries in Para-Swimming

In able-bodied freestyle swimming, 90% of the propulsion is from the upper limbs [64]. It is also a significant contributor to the propulsion in para-swimmers. Hand and forearm length are the most important factors in 100 m freestyle performance in para-swimmers [65]. Bilateral mean hand force and swimming velocity are highly correlated across para-swimming classifications, where para-swimmers exhibiting an asymmetrical anthropometric profile or more severe physical impairment typically generate lower forces and velocities [66,67]. For a unilateral arm-amputee swimmer, the reduced limb length and surface area will affect the ability to generate propulsive forces and reduce stroke length [66]. In addition, the lag time between the propulsive forces will require an increase in the stroke frequency to increase free swim velocity, increasing the swimmer’s overall mechanical work and energy cost to overcome the hydrodynamic drag [68]. Therefore, coaches should identify different technical skills to offset the asymmetric propulsive forces and asymmetric roll amplitude between the affected and unaffected sides [69].

### 4.4. Drag Factors

A physical impairment may affect the ability to generate propulsive forces to overcome the drag and inertial parameters in swimming. Payton et al. [70] found higher active and passive drag in para-swimmers with central motor and neuromuscular impairments when compared to able-bodied swimmers. Impairments in motor coordination or ROM predispose athletes to increased form drag, with para-swimmers exhibiting significantly reduced ROM compared to their able-bodied peers [71]. Unique head positions and postures may also affect the streamline position and thus increase form drag. Interestingly, para-swimmers with anthropometric impairments (limb deficiency, short stature, impaired ROM) show similar active and passive drag to non-disabled swimmers and swimmers from different sport classes [70]. This suggests that para-swimmers with anthropometric impairments are primarily limited by their ability to produce power rather than overcoming high active or passive drag. It is possible that these swimmers create less disturbance in the water with the increasing severity of impairment due to partial or full absence of a leg-kick or arm stroke during freestyle swimming. Although the lower limbs contribute less to propulsion than the upper body, the legs may still contribute to generating lift forces, thereby decreasing trunk inclination and form drag [72]. Para-swimmers with impairments to the lower limbs may need a greater emphasis on hip and leg strength, especially those in lower sport classes where drag plays a greater role on performance [73]. In response to understanding the importance of drag on athletes with varying impairments, World Para-Swimming has recently announced that drag will be one of the factors considered as part of the Functional Classification system process [70].

## 5. Conclusions and Future Directions

This narrative review critically evaluated the relevant biomechanical research published since 2016 and has expanded on the systematic review by Morriën [1] in two impactful ways to the Paralympic sport practitioner: we have summarized and critically evaluated the relevant Paralympic biomechanical literature from a practitioner-specific perspective. This review further highlights the athlete-specific biomechanical considerations required to optimize performance (Table 2). These specific considerations include disability-and event-specific training modalities and equipment configurations to optimize performance and reduce the incidence of injury. Future directions include the use of Paralympic sport biomechanics to contribute to the objective classification of Paralympic athletes and the examination of longitudinal changes in Paralympic biomechanics as a result of the systematic and well-documented training studies aimed at optimizing Paralympic performance.

## Figures and Tables

**Figure 1 sports-09-00089-f001:**
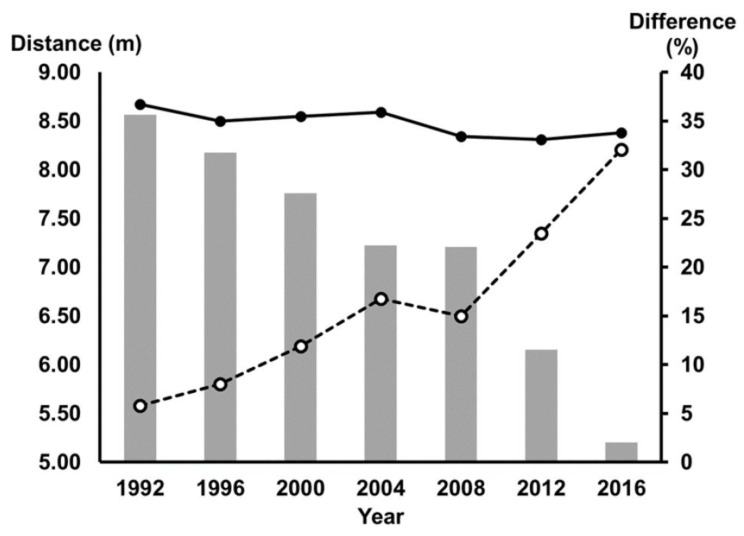
Men’s gold medal winning long jumps for the Olympics and Paralympic Games from 1992 to 2016. Olympic-winning jumps are shown in filled circles. Paralympic-winning F44 jumps are shown in open circles. Filled bars indicate the percent difference between Olympic and Paralympic winning jumps.

**Table 1 sports-09-00089-t001:** 2020(1) Paralympic sports, sport-specific body posture, and impairment(s).

SPORT	POSTURE	IMPAIRED	IMPAIRED	LIMB	LEG LENGTH	INTELLECTUAL	INVOLUNTARY	MUSCLE	UNCOORDONATED	SHORT	VISION
		**MUSCLE POWER**	**PASSIVE ROM**	**DEFICIENCY**	**DIFFERENCE**	**IMPAIRMENT**	**MOVEMENTS**	**TENSION**	**MOVEMENTS**	**STATURE**	**IMPAIRMENT**
**Archery**	STANDING SITTING	x	x	x				x	x		x
**Athletics**	STANDING	x	x	x	x	x	x	x	x	x	x
	SITTING										
**Badminton**	STANDING SITTING	x	x	x	x		x	x	x	x	
**Boccia**	SITTING	x	x	x			x	x	x		
**Canoe Sprint**	SITTING	x	x	x							
**Cycling**	SITTING	x	x	x	x		x	x	x		x
**Equestrian**	SITTING	x	x	x	x		x	x	x	x	x
**Football 5-a-side**	STANDING										x
**Goalball**	STANDING										x
**Judo**	STANDING										x
**Powerlifting**	SUPINE	x	x	x	x		x	x	x	x	
**Rowing**	SITTING	x	x	x			x	x	x		x
**Shooting**	STANDING SITTING	x	x	x			x	x	x		x
**Swimming**	SUPINE	x	x	x	x	x	x	x	x	x	x
**Table Tennis**	STANDING SITTING	x	x	x	x	x	x	x	x	x	
**Taekwondo**	STANDING	x		x	x	x	x	x	x		
**Triathlon**	STANDING SITTING	x	x	x			x	x	x		x
**Wheelchair Basketball**	SITTING	x	x	x	x		x	x	x		
**Wheelchair Fencing**	SITTING	x	x	x			x	x	x		
**Wheelchair Rugby**	SITTING	x	x	x			x	x	x		
**Wheelchair Tennis**	SITTING	x	x	x	x		x	x	x		

**Table 2 sports-09-00089-t002:** Practical and special consideration for supporting Paralympic athletes.

Key Intervention	Potential Impact of Intervention	Considerations, Benefits, Knowledge Gaps
RSP alignment relative to sagittal plane	▪ Increased step frequency	▪ Higher top and average running speed over 100 m.
	▪ Higher propulsive impulse	
RSP height	▪ Longer stride length	▪ Maximal allowable standing height regulated by IPC
	▪ Lower stride frequency	▪ CAS disallowed Blake Leeper to compete in 2020(1) Olympics based on RSP height
		▪ Moment of inertia of RSP dictates changes in stride frequency, which may be different than intact limb(s)
RSP stiffness	▪Higher energy return compared to compliant blade	▪ Energy “return” is always <100%, and less than intact limbs since RSPs cannot generate positive mechanical power
		▪ Energy storage and return does not incur a metabolic cost, like in intact limbs
Intensive and repetitive practice for those athletes with neurological impairments (reduced motor control)	▪ Improved consistency in movement execution, reduced movement variability and compensational movements.	▪ Intensive practice may improve muscle coordination and maximal voluntary activation in these individuals. Eg. Higher impulses over short time periods during the block phase will improve swim start performance
Use of stability and proprioceptive training for VI athletes	▪ Improved kinesthetic awareness, neuromuscular control, and orientation.	▪ VI Athletes may have subconscious fears to movement disruption or fear of injury during training and competition. Methods to help overcome these fears may lead to increased running speeds.
Strength Training for injury prevention in athletes with upper body predominant propulsion (WC sports, swimmers with LLA, etc.)	▪ Symmetrical force application between sides will increase power output.	▪ Athletes with a greater demand on particular limb(s) may show greater propulsion asymmetries and develop compensations for control. Coaches should identify different technical skills, strength trainers should identify strength imbalances to offset the asymmetric force application.
	▪ May reduce the chances of overuse injury	
Athlete-specific equipment configurations (wheelchair, boat, prosthesis, etc)	▪ Improvements to force application, propulsion efficiency, drag, range of motion and possibly injury prevention.	▪ Athlete-specific body configurations and equipment choices can be made from objective biomechanical inputs and lead to performance enhancement.

Evaluate sport-specific mobility in wheelchair athletes	▪ Quantify and monitor an athlete’s relative weaknesses and strengths, overall progress, recovery from injury, or improve classification objectivity.	▪ The tests need to be valid and reliable; discriminate between athletes of different sex, competition level, and level of impairment

Hip and knee extensor muscle strengthening in Paralympic sprinters, long jumpers, and para-swimmers with lower limb impairments	▪ Increased joint angular velocities, external joint powers and step length during the initial acceleration phase of sprinting.	▪ In CP, hip joint performs negative work due to increased passive and active joint stiffness.
	▪ Improvements to horizontal take-off velocities during the long jump	▪ In unilateral LLA long jumpers, specific eccentric strengthening of hip and knee extensors can help increase leg stiffness during the pivot.

	▪ Improvements in hip and leg strength may contribute to generating lift forces, and thereby decreasing trunk inclination and form drag in Para-swimmers	▪ Drag can differ appreciably between athletes of different sport classes. An improved streamline position will reduce drag and improve performance.

Abbreviations: Court of Arbitration for Sport (CAS); Cerebral Palsy (CP); Lower leg amputee (LLA); Running-specific prosthetic (RSP); Visual impairment (VI).

## Data Availability

Data sharing not applicable. No new data were created or analyzed in this study. Data sharing is not applicable to this article.

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
