# Peer review of "How Can Biomechanics Improve Physical Preparation and Performance in Paralympic Athletes? A Narrative Review"

_sports, 2021, doi:10.3390/sports9070089_

Round 1
Reviewer 1 Report
Review paper containing a lot of valuable information for readers interested in the sport development in the field of paralympic athletes. In fact, proper hardware configuration improves the results, and the authors objectively provided evidence to support this assumption. I recommend this article to publication.
Author Response
Thank you for the encouraging comments regarding our manuscript. We hope the suggested revisions further to improve the manuscript.
Reviewer 2 Report
In this paper, the authors summarize and review recently relative research articles for preparation and performance analysis for Paralympic athletes. They provide biomechanical training issues and equipment configuration issues for Paralympic sports.
please see attachment for detailed comments.

Author Response
We want to sincerely thank the three reviewers their time in serving as referee for this manuscript and for the valuable feedback regarding our submitted narrative review. We hope to have adequately addressed each of these reviewer’s comments below. We feel the manuscript is substantially improved as a result of these peer-reviewers feedback.
We have addressed the specific comments below in bold.
Referee 2
How can biomechanics improve physical preparation and performance in Paralympic
athletes? A narrative review
In this paper, the authors systemically narrative review relevant works for Paralympic
preparation and performance analysis. They completely summarize articles published
from 2007 to 2016.
This paper is a review article. Comparing to research article, there is less originality.
Most of materials in this paper are relevant works and the authors’ summarization
comments. However, from review article, this paper provides complete
summarization of relevant works. The authors classify Paralympic into standing,
jumping, sitting and horizontal four categories. And they further analysis the
parameters for each other. Coaches and athletes can refer to this narrative
description suggests coaches and athletes how to training and adjusting equipment
for Paralympic.
This paper review the relevant articles published between 2007 and 2016. From the
review article, the relevant works in this paper is too old. Although the Paralympic
2020 extending to 2021, however, the research works still be published. The authors
should provide latest works to provide new information for readers.
The English writing is well, and grammar is correct. The authors summarize 50
articles in short words and further describe with tables. Readers can easily and
clearly read and get the information from this paper. However, the spatial resolution
of Figure 1 is poor. In case of the authors have already get permission, I suggest that
Figure 1 can be re-painted with higher resolution. Otherwise, I suppose this figure
can be removed.
Thank you for this valuable feedback. Based on these suggestions above, and from the other two reviewers, Figure 1 has been removed.
To address the timeliness of the relevant articles, respectively, we have cited many relevant studies since 2017 (eg. references #1, 8, 9, 15, 18, 19, 26, 30, 31, 32, 33, 34. 38, 39, 42, 43, 46. 47, 48 and 51). We simply use older studies to support/contradict these studies in our narrative review. Also, while we have not presented a systematic review of Paralympic biomechanics, we believe we have included all relevant studies since 2016 into this narrative review. If the referee feels we may have missed relevant literature since 2017 in which to critique/comment, we would be happy to consider these.
Reviewer 3 Report
This study aimed at summarizing the aimed at providing an update of knowledge regarding Paralympic athletes from a biomechanical perspective. Even though the topic is interesting and relevant, evidence is not summarized using a very high scientific and academic style. I do not feel enthusiastic in reading this review.
Specific comment
What do authors mean with “to critically evaluate”?
Line 50. Please be more specific with the range period of the previous systematic review.
Line 51. What do you refer with “evidence-based classification”?
Line 54. What do you refer to “performance biomechanics”?
Line 74-76. This sentence requires a reference. Please be consistent with the use of stride or step.
Why are these two paragraphs with bullets?
Authors used abbreviations which are not very common in literature, making difficult to follow the text.
Space between number and metric unit is necessary
Quality of table 2 should be increased.
Reference list has to follow the journal guidelines.
Author Response
We want to sincerely thank the three reviewers their time in serving as referee for this manuscript and for the valuable feedback regarding our submitted narrative review. We hope to have adequately addressed each of these reviewer’s comments below. We feel the manuscript is substantially improved as a result of these peer-reviewers feedback.
We have addressed the specific comments below in bold.
Referee 3:
This study aimed at summarizing the aimed at providing an update of knowledge regarding Paralympic athletes from a biomechanical perspective. Even though the topic is interesting and relevant, evidence is not summarized using a very high scientific and academic style. I do not feel enthusiastic in reading this review.
Specific comment
What do authors mean with “to critically evaluate”?
We have clarified this vague statement to read: “Here, we serve to update this review, and to examine the impact specific biomechanical interventions may have on Paralympic performance”
Line 50. Please be more specific with the range period of the previous systematic review.
We have stated that Morrien conducted a systematic review to include all Paralympic biomechanics studies published prior to July, 2016.
Line 51. What do you refer with “evidence-based classification”?
Evidence-based classification was one of three themes highlighted by the previous systematic review by Morrien et al. (2017). This included six studies examining evidence-based classification in Paralympic sport (Morrien’s references 5,6,10,11,17,32). Note: some of these references are included, and critically evaluated from a performance improvement perspective, in the current narrative review .
Line 54. What do you refer to “performance biomechanics”?
This sentence has been removed. Thank you.
Line 74-76. This sentence requires a reference. Please be consistent with the use of stride or step.
Thank you. A reference has been included with this statement. For clarity, we have revised all reference to stride (2 steps) length or frequency to step length or frequency, respectively.
Why are these two paragraphs with bullets?
These summary points of our narrative review have been removed. Thank you.
Authors used abbreviations which are not very common in literature, making difficult to follow the text.
Thank you. We have removed all non-common abbreviations from the manuscript text. We hope this serves to improve the readability of the narrative review.
Space between number and metric unit is necessary
This has been corrected in the revised manuscript. Thank you.
Quality of table 2 should be increased.
Thank you for this suggestion. We have increased the quality and font size of Table 2 in the revised submission.
Reference list has to follow the journal guidelines.
Respectively, we note from the author guidelines of the journal (https://www.mdpi.com/journal/sports/instructions) that referencing may be in any format style. Accordingly, we have used a common reference style, formatted with Mendeley as recommended by the author guidelines webpage. If the journal now has a preferred reference style, which is not reflected in the author instructions, we would be happy to re-format to that style.
Round 2
Reviewer 3 Report
Authors provided their best effort to improve the quality of manuscript.